# Inflammation induced by incomplete radiofrequency ablation accelerates tumor progression and hinders PD-1 immunotherapy

Liangrong Shi [1,2], Junjun Wang[3], Nianhua Ding[1,2], Yi Zhang[4], Yibei Zhu[5], Shunli Dong[4], Xiaohui Wang[4], Changli Peng[1], Chunhui Zhou[1], Ledu Zhou[6], Xiaodong Li[3], Hongbing Shi[3], Wei Wu[7], Xueyin Long[1,2], Changping Wu[3]* & Weihua Liao[1,2]*

Radiofrequency ablation (RFA) promotes tumor antigen-specific T cell responses and enhances the effect of immunotherapy in preclinical settings. Here we report that the existence of remnant tumor masses due to incomplete RFA (iRFA) is associated with earlier new metastases and poor survival in patients with colorectal cancer liver metastases (CRCLM). Using mouse models, we demonstrate that iRFA promotes tumor progression and hinders the efficacy of anti-PD-1 therapy. Immune analysis reveals that iRFA induces sustained local inflammation with predominant myeloid suppressor cells, which inhibit T cell function in tumors. Mechanistically, tumor cell-derived CCL2 is critical for the accumulation of monocytes and tumor-associated macrophages (TAMs). The crosstalk between TAMs and tumor cells enhances the CCL2 production by tumor cells. Furthermore, we find that administration of a CCR2 antagonist or the loss of CCL2 expression in tumor cells enhances the antitumor activity of PD-1 blockade, providing a salvage alternative for residual tumors after iRFA.

[1] Radiological Intervention Center, Department of Radiology, Xiangya Hospital, Central South University, Changsha 410005 Hunan, China. [2] Center for Molecular Imaging, Xiangya Hospital, Central South University, Changsha 410005 Hunan, China. [3] Department of Oncology, the Third Affiliated Hospital, Soochow University, Changzhou 213003 Jiangsu, China. [4] Dept. of Pharmacology, College of Pharmaceutical Sciences, Soochow University, Suzhou 215123 Jiangsu, China. [5] Institute of Biotechnology, Key Laboratory of Clinical Immunology of Jiangsu Province, Soochow University, Suzhou 215123 Jiangsu, China. [6] Department of General Surgery, Xiangya Hospital, Central South University, Changsha 410005 Hunan, China. [7] National Clinical Research Center for Geriatric Disorder, Xiangya Hospital, Central South University, Changsha 410005 Hunan, China. *email: wcpjjt@163.com; owenliao@csu.edu.cn

The PD-L1 (programmed death ligand 1)/PD-1 block-ade immunotherapy induces long-lasting responses and prolongs overall survival (OS) in patients with metastatic cancers[1]. However, a majority of patients do not benefit from this treatment[2]. The efficacy of anti-PD-L1/PD-1 therapy is closely associated with pre-existing anti-tumor immune responses[3–5]. Thus, the rational combination of treatments that can induce anti-tumor immune responses and the anti-PD-L1/PD-1 therapy should produce synergistic efficacy for a larger number of cancer patients. Local treatment with thermal ablation cannot only reduce the tumor burden but also stimulate the host anti-tumor immune response, thereby showing a promising prospect of being combined with checkpoint blockade immunotherapy (CBI).

The thermal ablation of tumors is the local application of extreme temperatures to induce coagulative necrosis for the cure or palliation of many tumor types as an alternative to surgery[6]. Radiofrequency ablation (RFA) is the most commonly used thermal technique. In fact, RFA combined with CBI has been proven to be effective in preclinical and some clinical studies[7–11]. However, RFA has also been reported to cause rapid tumor progression[12–17], which might be due to incomplete ablation of the target tumors[14,18]. The present study aims to evaluate the immune response in residual tumors after incomplete RFA (iRFA) and explore whether the combination of RFA and immunotherapy can be extended to patients with advanced tumors that cannot be ablated completely.

## Results

**iRFA is associated with poor survival in CRCLM patients**. In order to study whether the presence of the residual tumor is associated with rapid disease progression after iRFA, we performed a retrospective case-controlled study of patients with CRCLM. From January 2008 to December 2016, we treated 551 patients with CRCLM using RFA. Among them, 43 patients were found to have incomplete ablation, which was defined as observable residual tumor within or at the periphery of the original ablative zone within 4 months after RFA (Group A). As the control group (Group B), 43/506 patients, age- and sex-matched but without local residual tumor after RFA during the same study period were selected. The clinicopathological characteristics of the enrolled patients are listed in Supplementary Table 1. In Group A, the main risk factors related to observable residual tumor included the size of tumor greater than 3 cm, contact with portal vein and hepatic hilum, and adjacency to gastrointestinal tract (Supplementary Table 2). Typical CT images of a patient with local residual tumor and new hepatic metastasis are shown in Fig. 1a. Next, we compared the time to new metastasis (TTNM) (including extrahepatic and intrahepatic metastases separated from the RFA zone) and OS between groups A and B.

Significantly shorter TTNM and OS were observed in group A comparing to the group B (Fig. 1b, c). Multivariate analysis identified incomplete ablation as an independent risk factor for the earlier occurrence of new metastasis and poor survival (Supplementary Table 3).

**iRFA accelerates tumor progression and hinders PD-1 therapy**. To further study the immune reaction in the residual tumor, mice were inoculated with colon cancer cell lines CT26 and MC38, which can be inhibited by the RFA-elicited immune response[7,19]. Next, iRFA was performed to cause partial necrosis of tumors (Fig. 2a). We found that iRFA initially reduced the size of the treated tumor. However, the residual tumors eventually grew even larger than the control tumors (Fig. 2b, c). Notably, a greater number of metastasis was found in the cavity, liver, and lung at day 14 post-iRFA (Fig. 2d). The survival time was significantly reduced in mice treated with iRFA as comparing to the untreated mice (Fig. 2e). Next, we sought to determine whether the thermal injury in the normal tissue caused by RFA leads to tumor progression. RFA was performed in subcutaneous tissue at the contralateral flank ensuring no damage to the implanted tumor (Supplementary Fig. 1A). However, no significant difference in tumor growth and distant metastasis between RFA of normal tissues and control groups was observed (Supplementary Fig. 1B). Collectively, these data suggest that the presence of residual tumor causes rapid disease progression after iRFA.

We have shown previously that PD-1 blockade enhances the RFA-elicited antitumor immunity, and hence, we sought to investigate whether the PD-1 blockade therapy inhibited the growth of the residual tumor after iRFA (Fig. 2f). Treatment with PD-1 mAbs alone could inhibit the growth of CT26 and MC38 tumor. However, the PD-1 blockade therapy had little effect in combination with iRFA; no significant differences were detected in the growth curve of residual tumor, the number of metastasis, and survival of mice between iRFA plus PD-1 mAbs and the iRFA groups (Fig. 2g–j). Futher, we confirmed this finding in mice bearing Hepa1-6 tumors, a hepatic cancer model that responds to PD-1 blockade therapy[20] (Supplementary Fig. 2A–D). These data suggested that residual tumors after iRFA are resistant to anti-PD-1 therapy.

**iRFA elicits inflammation and induces immune suppression**. To obtain an unbiased understanding of the underlying molecular mechanism of iRFA, we performed RNA-seq analysis using fresh residual CT26 tumors 3 days after iRFA as well as control tumors. Our analysis revealed distinct gene expression profiles between the iRFA-treated tumor and the untreated tumor (Fig. 3a, b). Gene Ontology (GO) analysis of differentially expressed genes showed

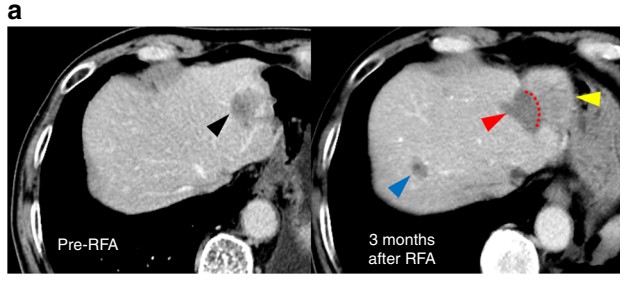

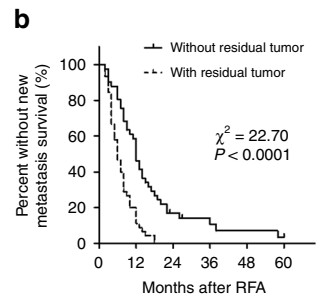

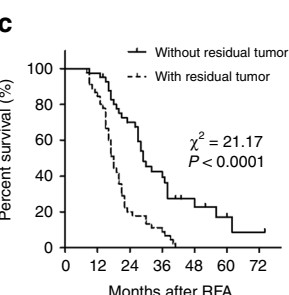

**Fig. 1** iRFA is associated with earlier new metastasis and poor survival in CRCLM patients. A retrospective study of patients with CRCLM treated with RFA, in which 43 patients with local residual tumors were divided into Group A and another 43 matched patients without local residual were divided into Group B. **a** Typical CT images of a patient who had local residual tumor and new hepatic metastasis (black arrow: target tumor pre-RFA, red arrow: necrotic zone, yellow arrow: local recurrent tumor, blue arrow: new hepatic metastasis ($n = 43$); **b** Kaplan–Meier curves of time to new metastasis (TTNM); **c** Kaplan–Meier curves of overall survival (OS) ($n = 43$). Source data are provided as a Source Data file.

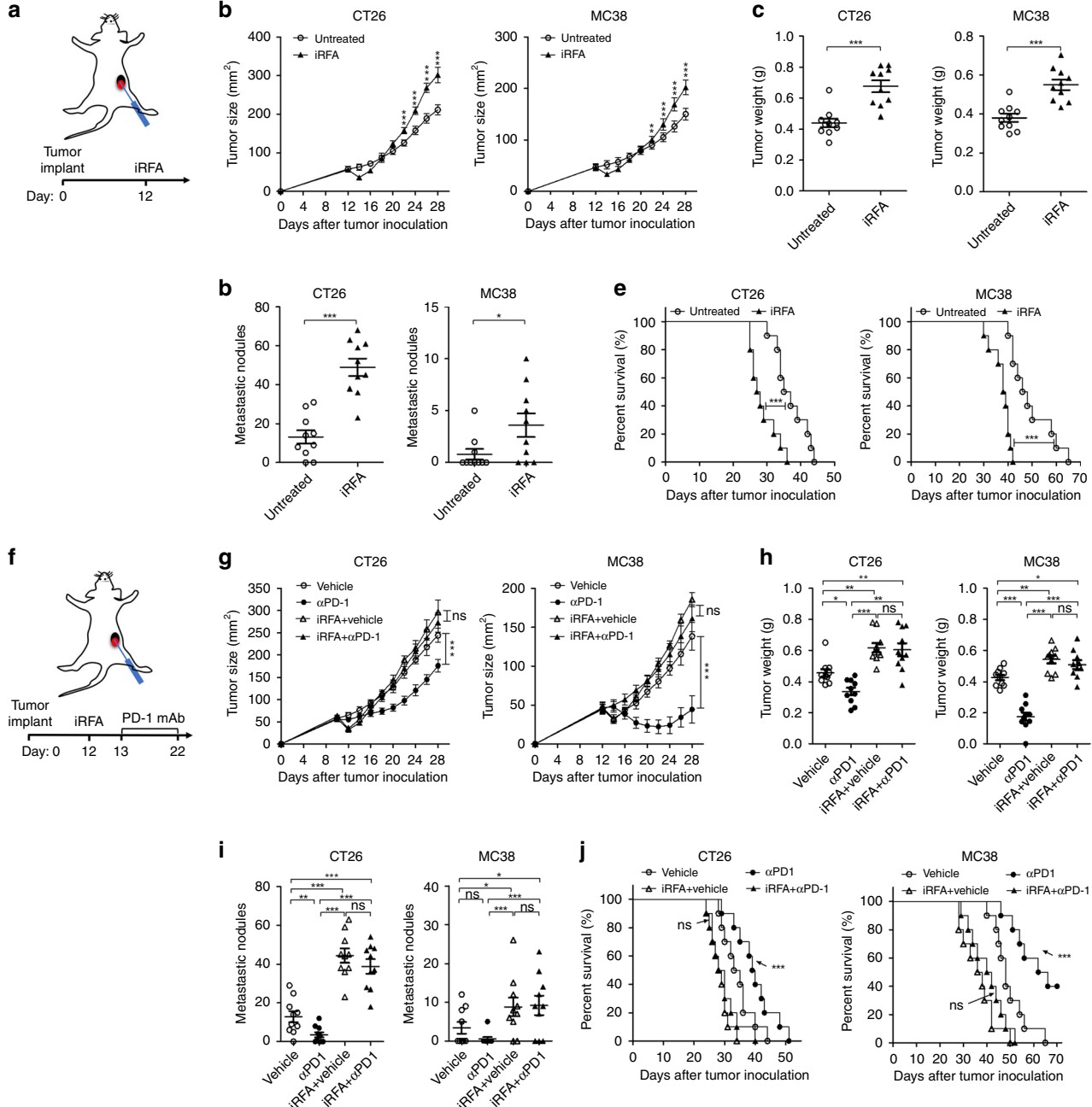

**Fig. 2** iRFA promotes tumor progression and hinders anti-PD-1 therapy. **a.** Diagram of iRFA treatment. $1 \times 10^6$ CT26 or MC38 cells were injected *i.d.* into male BALB/C and C57BL/6 mice, respectively, on right flank. iRFA was performed to cause partial necrosis of the tumors when the longest dimension reach about 0.8 cm. **b** Growth curve of the residual tumor ($n = 10$). **c**. Weight of the residual tumor examined on day 14 after iRFA by dissection the mice ($n = 10$). **d** Number of the metastasis examined on day 14 after iRFA by dissection the mice ($n = 10$). **e** Kaplan–Meier survival curves are shown and the log-rank test was performed ($n = 10$). **f** Schematic of the study of combined therapy with iRFA and PD-1 inhibition. Anti-PD-1 mAb (200 μg, clone: J43) or vehicle was administered through intraperitoneal injection to mice every 3 days for a total of four times. **g** Growth curve of the residual tumor (one-sided ANOVA test, ns presents not significant, ***$P < 0.001$, $n = 10$). **h** Weight of the residual tumor examined on day 14 after iRFA by dissection the mice. **i** Number of metastasis examined on day 14 ($n = 10$). **j** Kaplan–Meier survival curves are shown and the log-rank test was performed (ns presents not significant: compared to iRFA + vehicle, ***$P < 0.001$: compared to the other three groups, $n = 10$). Data represent cumulative results from 1/2 independent experiments with 10 mice per group. Statistical differences between pairs of groups were determined by a two-tailed Student's *t*-test. (ns presents not significant, *$P < 0.05$, **$P < 0.01$ ***$P < 0.001$). Each error bar represents means ± SEM. Source data are provided as a Source Data file.

that the top enriched terms included inflammatory response, immune response, immune system processing, leukocyte chemotaxis and migration, hypoxia, CCR2 chemokine receptor binding, and response to wounding (Fig. 3c). Genes encoding damage-associated molecular patterns molecules (DAMPs), proinflammatory cytokines, chemokines, and genes associated with

immunosuppression were overexpressed in the residual tumor (Fig. 3d). These data indicate that iRFA elicited complex inflammatory reaction in residual tumor, which ultimately promotes immunosuppression in the tumor microenvironment (TME).

In order to further study the kinetics of inflammation, we focused on TNFα, an important proinflammatory cytokine[21]

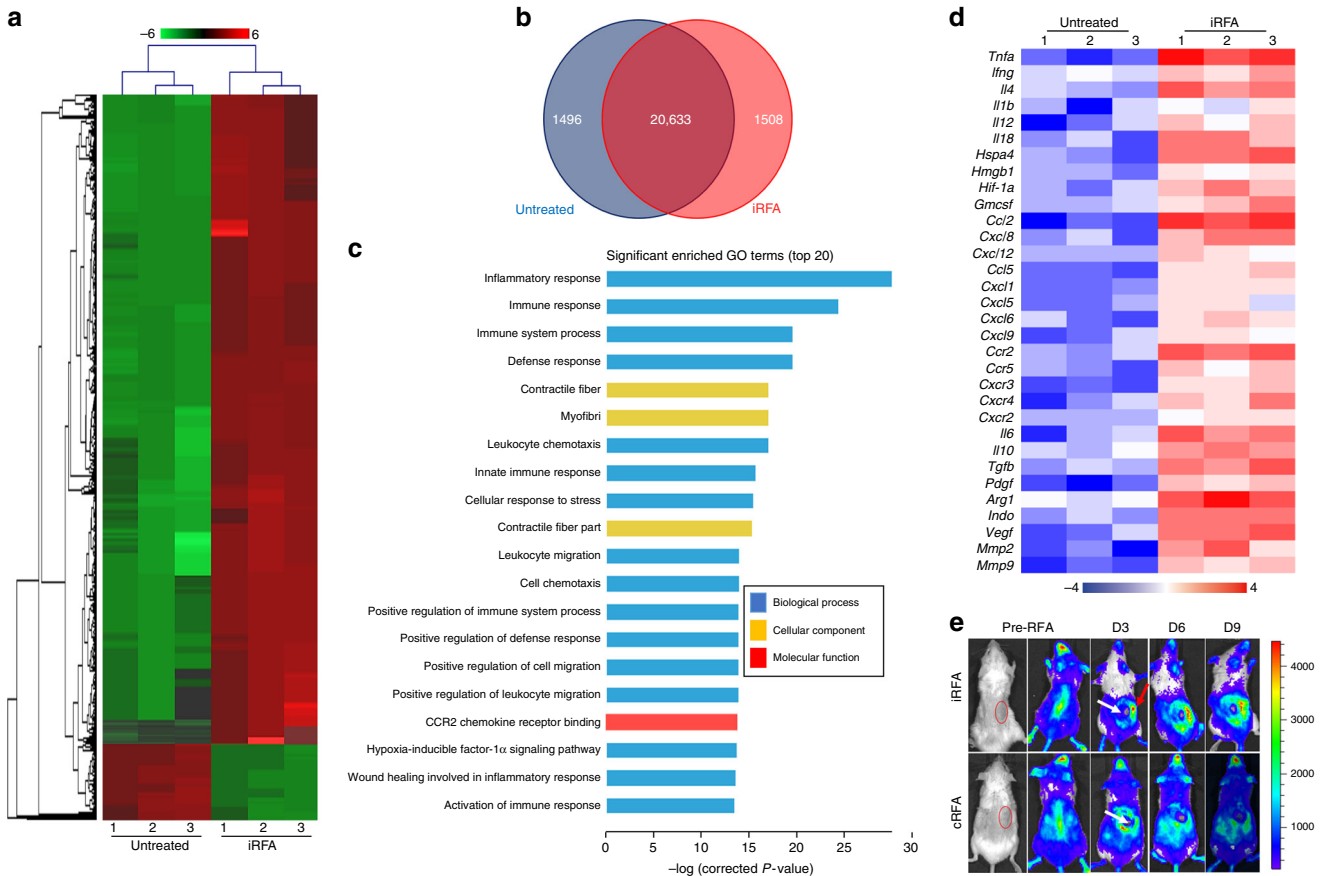

**Fig. 3** iRFA induces local inflammation and upregulates genes associated with immune suppression. **a** Difference in RNAseq of the untreated and iRFA-treated CT26 tumors on day 3 after iRFA ($n = 3$). Red and green colors indicate upregulated or downregulated genes. **b** Overlap in gene expression profiles between untreated and iRFA-treated tumors ($n = 3$). **c** Significant enrichment in GO terms (top 20). **d** Heatmap of mean fold-change of in cytokines and chemokines and genes-associated immunosuppression ($n = 3$). Red and blue colors indicate upregulated or downregulated genes. **e** Bioluminescence imaging of BALB/C-Tg (Tnfa-luc)-Xen mice inoculated with complete or incomplete ablation of the implanted CT26 tumors (red circle: targeted tumor pre-RFA white arrow: necrotic zone, red arrow: residual tumor). Red and blue represent high or low fluorescence intensity ($n = 3$).

highly upregulated in the residual tumor as comparing to the untreated tumor (Fig. 3d). BALB/C-Tg (TNFα-luc)-Xen mice were inoculated with CT26 cells and treated with iRFA or complete RFA (cRFA). On day 3, we observed a high TNFα expression in the adjacent tissue around the treatment zone. Subsequently on day 9, this signal weakened and finally disappeared along with wound healing in mice receiving cRFA. Notably, we observed a sustained high signal of TNFα expression in the residual tumor 9 days after iRFA (Fig. 3e). This result indicates that RFA-induced local inflammation persists longer after iRFA comparing to cRFA.

**Infiltrating myeloid cells increase in residual tumor**. To better understand the cellular mechanisms underlying iRFA-induced inflammation, we analyzed the infiltrating immune cells in residual CT26 and MC38 tumors in comparison to those in untreated tumors. An earlier (day 3) and later time point (day 9 when inflammation subsided in adjacent normal tissue after cRFA with wound-healing) were selected to represent the acute and chronic stages, respectively. On day 3, a strong inflammatory reaction with more than a three-fold increase in CD45$^+$ immune cells was observed in the residual tumor (Fig. 4a, Supplementary Fig. 3A). This increase of immune cells was primarily attributed to infiltrating CD11b$^+$ myeloid cells as both the absolute number and the relative percentage of these cells in CD45$^+$ cells were significantly

increased (Fig. 4a, Supplementary Fig. 3A). Furthermore, we subdivided myeloid cells into neutrophilic (CD11b$^+$Ly6G$^{hi}$) myeloid derived suppressor cells (MDSC), monocytic (CD11b$^+$Ly6C$^{hi}$) MDSC, and macrophages (CD11b$^+$F4/80$^+$). All these myeloid cell subsets showed a substantial increase in absolute number and percentage in CD45$^+$ cells (Fig. 4a, Supplementary Fig. 3A). Simultaneously, we observed that nMDSC and mMDSC were also increased in the peripheral blood, indicating a systemic inflammatory response after iRFA. On day 9, the number of peripheral nMDSC and mMDSC decreased closely to the level of untreated mice (Supplementary Fig. 4A, B, Supplementary Fig. 5A). Interestingly, we still observed a sustained increase in myeloid cell infiltration in the residual tumor on day 9. On day 9, the proportion of nMDSC in CD45$^+$ cells declined by half, but the infiltration of mMDSC and macrophages still maintained a high level (Fig. 4a, Supplementary Fig. 3A). We further quantified the tumoral inflammatory M1-like (CD11b$^+$F4/80$^+$MHCII$^+$) and immuno-suppressive M2-like (CD11b$^+$F4/80$^+$CD206$^+$) macrophage subsets. A relatively higher proportion of M2-like macrophages were observed in the residual tumor as comparing to the control untreated tumor on day 3, and a later stage, the macrophages were further shifted to the M2 phenotype (Fig. 4b, Supplementary Fig. 3B). Consecutively, the mRNA expression of M2 markers (TGFβ, Arg-1, IDO) was elevated (Fig. 4c), and the concentration of immunosuppressive cytokines IL-10 and TGFβ increased in peripheral blood (Supplementary Fig. 5B). The expression of PD-L1

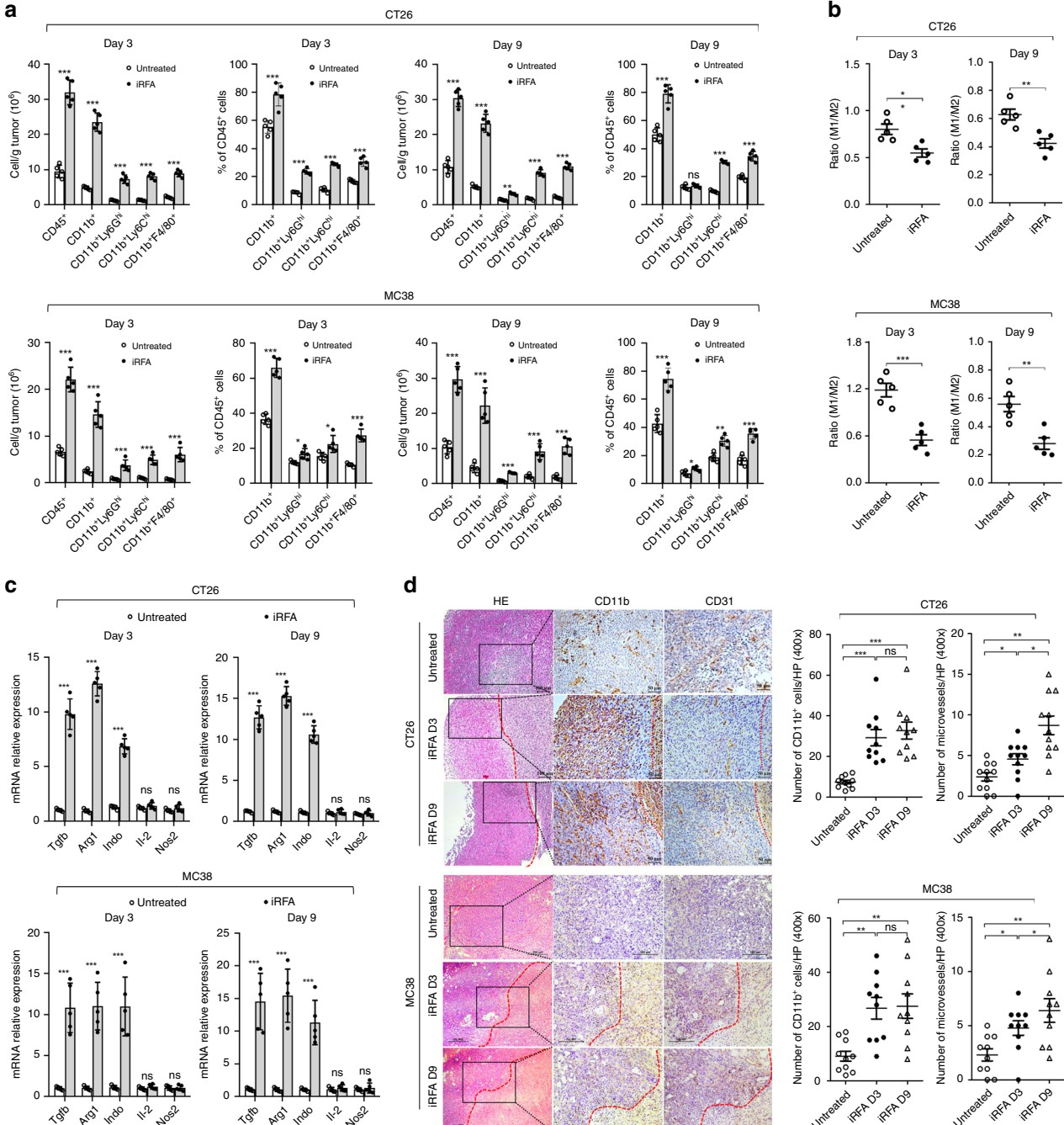

**Fig. 4** iRFA induces immunosuppressive myeloid cells accumulation in the residual tumors. iRFA was administrated in CT26 and MC38-bearing mice as described in Fig. 1. On days 3 and 9 after iRFA treatment, the residual tumors were resected and either digested to generate single cell suspension or to isolate RNA or embedded in paraffin. **a** Flow cytometric quantification of CD11b+, CD11b+Ly6G+, CD11b+Ly6C+, and CD11b+F4/80+ cells (gate on CD45+ liver cells) in the untreated and iRFA-treated CT26 and MC38 tumors ($n = 5$). **b** Flow cytometric quantification of CD11b+F4/80+ (TAM) population, and expression of CD206 and MHCII in CD11b+F4/80+ cell populations in the untreated and iRFA-treated CT26 and MC38 tumors ($n = 5$). **c** mRNA expression of selected M1 and M2 markers as determined by qRT-PCR, data were showed as fold change ($n = 5$). **d** Representative microphotographs are showing H&E, CD11b, and CD31 staining of the untreated and residual CT 26 and MC38 tumor (the necrotic ablation area is on the right side of the red dotted line). Scale bar = 50 μm (CT26), 100 μm (MC38). The data represent the number of positive cells in 10 different microscopical fields (400×) in 1 of 3 replicates. Data represent cumulative results from 1/2 independent experiments. The data are represented as mean ± SEM. Statistical differences between pairs of groups were determined by a two-tailed Student's t-test (ns present not significant, **$P < 0.01$, ***$P < 0.001$). Source data are provided as a Source Data file.

expression, however, was not altered significantly in the macrophages in the iRFA tumor (Supplementary Fig. 6A, B). Using immunohistochemistry (IHC) staining results revealed a diffused infiltration of CD11b+ cells in both residual tumor and necrotic area (Fig. 4d). In addition, we observed greater numbers of

microvessel as identified by CD31 staining in residual tumor when compared to untreated tumors (Fig. 4d). To identify whether the inhibitory effects are a result of RFA in general, we compared peripheral MDSCs and immunosuppressive cytokines after iRFA and cRFA. We found the number of CD11b+Gr1+ MDSCs was

reduced in mice treated with cRFA as comparing to iRFA and untreated mice (Supplementary Fig. 5A). The concentration of IL-10 and TGFβ also decreased in cRFA-treated mice on day 9 (Supplementary Fig. 5B).

**Infiltrating myeloid cells inhibits T cell functionality**. Next, we analyzed the number and function of T cells in the residual CT26 and MC38 tumor after iRFA. We observed that the ratio of CD3+ T to CD11b+ cells was significantly decreased in iRFA tumors comparing to untreated tumors (Fig. 5a, Supplementary Fig. 3C). The percentage of CD4+ and CD8+ T cells in CD45+ cells was also reduced in the residual tumor. In addition, the percentage of Treg increased and the CD8+/Treg (CD4+FoxP3+) ratio was reduced in the residual tumor (Fig. 5b, Supplementary Fig. 3D). Moreover, percentages of CD8+ cells producing granzyme B and IFNγ in tumors were lower in iRFA tumors comparing to control tumors (Fig. 5c, Supplementary Fig. 3E). Interestingly, the proportion of PD-1+CD8+ decreased in the residual tumor, consistent with a reduction in T cell responses after iRFA (Fig. 5c, Supplementary Fig. 3E). Furthermore, we examined the suppressive function of CD11b+ myeloid cells isolated from tumors and spleens of iRFA-treated mice. tumor-derived CD11b+ cells significantly inhibited the proliferation of naive CD8+ T cells in vitro (Fig. 5d). Subsequently, the isolated myeloid cells were mixed with tumor cells and adoptively transferred into the recipient mice (Fig. 5e). This treatment promoted tumor growth (Fig. 5f), and resulted in lower percentages of tumoral CD8+ cells and granzyme B+ and IFNγ+CD8+ T cells (Fig. 5g). Taken together, these data demonstrated that tumor infiltrating myeloid cells from iRFA-treated tumors had strong immune suppressive activity.

**CCL2 is critical for the accumulation of monocytes and TAMs**. In order to explain the underlying mechanism of myeloid cell recruitment, we focused on chemokine genes that were specifically upregulated after iRFA treatment. Analysis of RNA-seq data revealed that several chemokines that can facilitate the recruitment of myeloid cells, including CCL2, CXCL8, CCL5, and CXCL12, were upregulated on day 3 after iRFA (Fig. 3d). Among them, CCL2 was continuously overexpressed, while the expression of other chemokines decreased significantly at the later stage both in residual CT26 and MC38 tumor (Fig. 6a, Supplementary Fig. 7a, b). Correspondingly, CCL2 protein was increased in peripheral blood after iRFA. In contrast, peripheral CCL2 significantly decreased in mice treated with cRFA as comparing to iRFA and untreated-mice, suggesting the increase of CCL2 is originated from the residual tumor (Supplementary Fig. 5C). Furthermore, IHC confirmed that CCL2 was overexpressed in residual tumor cells adjacent to the necrotic zone with a large number of F4/80+ cell infiltration (Fig. 6b). CCL2 is a critical chemokine important for the recruitment of tumor-associated macrophages (TAMs)[22,23]. Therefore, we investigated whether CCL2 secreted by tumor cells was associated with the activity of myeloid cells. First, we observed that >70% of total myeloid cells and 90% of monocytes (CD11b+L6yC+) expressed CCL2 cognate receptor (CCR2) (Fig. 6c). Next, we knocked out the CCL2 gene by CRISP/Cas9 editing technique in CT26 and MC38 cancer cells. Consequently, the loss of CCL2 expression in cancer cells significantly decreased the total myeloid cells, monocytes and macrophages infiltration but not neutrophils (Fig. 6d). Furthermore, we observed CCL2 knockout downregulated expression of the immunosuppression-related genes (Fig. 6e) and attenuated the growth of the residual tumor (Fig. 6f). Less distant metastases were found in mice with CCL2−/− tumors (Fig. 6g). Taken together, tumor cells derived-CCL2 signal plays a vital role in

promoting the infiltration of monocytes and differentiation into TAMs, resulting in high invasiveness of the residual tumor.

**TAMs promote the CCL2 production by tumor cells through TNFα**. TNFα is a critical macrophage-produced mediator that elevates the production of CCL2 by multiple types of cancer cells[24,25]. In the residual tumor, the overexpression region of CCL2 and TNFα overlapped closely (Fig. 6b). The blockade of TNFα via a neutralizing antibody could downregulate the CCL2 expression both in residual CT26 and MC38 tumors (Fig. 7a, b). In order to explore the direct interaction between TAMs and cancer cells, CD11b+F4/80+ cells were isolated from residual tumors and co-cultured with CT26 and MC38 cancer cells in vitro, respectively. The level of CCL2 protein in suspension was increased in the co-cultured group (Fig. 7c). Then, neutralizing TNFα in the co-culture or pretreatment of cancer cells with anti-TNFR1 mAb significantly decreased the production of CCL2 (Fig. 7d). Next, we confirmed that TNFα stimulated the CCL2 production in a time-depended manner (Fig. 7e, f). These results indicated that TAMs promoted the CCL2 production by tumor cells through TNFα, which consequently forms a positive feedback loop, sustaining the infiltration of myeloid cells and inhibiting T cell activity in residual tumors (Fig. 7g).

**CCL2/CCR2 blockade enhances anti-tumor immunity after iRFA**. Since CCL2 was critical for myeloid cells accumulation in residual tumor, we assessed whether targeting myeloid cells with CCR2 antagonist (CCR2a) could inhibit the growth of the residual tumor. As the residual tumor was resistant to anti-PD-1 therapy, we also attempted to determine whether CCR2a treatment could overcome resistance to PD-1 immunotherapy. CCR2a and anti-PD-1 mAb were administrated alone or together after iRFA. Although CCR2a alone exerted little effect on tumor regression in intact CT26 tumors (Supplementary Fig. 8A, B), it could slow down the progression of residual CT26 tumors. Notably, dual-therapy with CCR2a and anti-PD-1 greatly inhibited the growth of residual tumors and reduced the distant metastases (Fig. 8a–c). The survival of mice receiving dual-therapy was significantly prolonged comparing to CCR2a and anti-PD-1 alone (Fig. 8d). Similar findings were revealed in MC38 colon cancer model (Fig. 8a–d). In addition, these findings were confirmed in another mouse model with hepa1-6 tumor, which also, recruited the suppressive myeloid by secretion of CCL2[26] (Supplementary Fig. 9A-D). The analysis of infiltrating lymphocytes in residual CT26 and MC38 tumors demonstrated that the number and cytolytic function of CD8+ cells were significantly increased in mice with dual-therapy (Fig. 8e, f). Further, we showed anti-PD-1 mAb was effective to inhibit the growth of the residual CCL2 knock out tumor. The survival time of mice bearing CCL2−/− CT26 and MC38 was prolonged by PD-1 therapy in the setting of iRFA (Fig. 8g, h). Collectively, these data suggested that targeting CCL2/CCR2 enhanced anti-tumor immunity in the residual tumor, thereby overcoming the resistance to immune checkpoint blockade therapy.

**Discussion**
Increasing evidence have shown that the combination of thermal ablation and immunotherapy is a promising strategy for cancers[7,14,27−29]. Herein, we reported that incomplete ablation of a target tumor limited the efficacy of anti-PD1 immunotherapy. First, we found that the local residual tumor was an independent risk factor for the earlier new metastases and poor survival in CRCLM patients treated with RFA. Thus, mouse models were utilized to demonstrate that iRFA accelerated the tumor progression and hindered the PD-1 blockade immunotherapy.

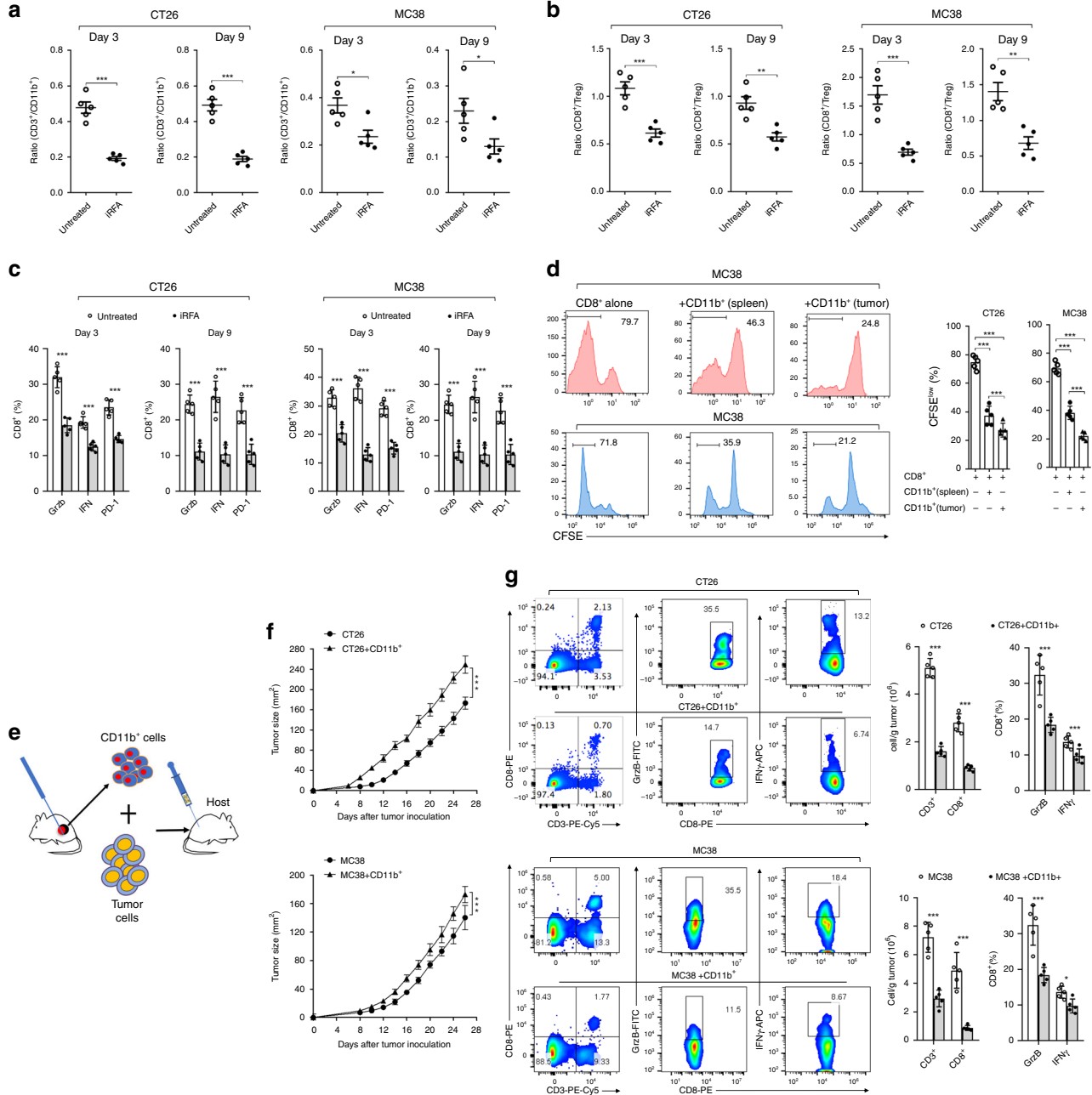

**Fig. 5** Infiltrating myeloid cells inhibit T cell functionality in residual tumors. **a** Flow cytometric analysis of CD11b⁺, CD3⁺ and the ratio of CD3⁺/CD11b⁺ in the untreated and iRFA-treated CT26 and MC38 tumors on day 3 and day 9 ($n = 5$). **b** Flow cytometric analysis of CD8⁺ and CD4⁺FoxP3⁺ cells and the ratio of CD8⁺/CD4⁺FoxP3⁺ in the untreated and iRFA-treated CT26 and MC38 tumors on day 3 and day 9 ($n = 5$). **c** Flow cytometric analysis and quantification of granzyme B, IFN-γ and PD-1 expression in CD8⁺ cells in the untreated and iRFA-treated CT26 and MC38 tumors on day 3 and day 9 ($n = 5$). **d** In vitro suppressive activity of tumor-infiltrating CD11b⁺ cells purified from spleen or CT26 residual tumors on day 3 after iRFA. Representative histograms of CD8⁺ T cell proliferation at a ratio of 1:1 CD8⁺ to CD11b⁺ T cells and percent CD8⁺ T cell proliferation ($n = 5$). **e-g** CD11b⁺ cells isolated from the residual tumor were mixed with CT26 or MC38 tumor cells and transferred into recipient mice. **e** Adoptive transfer method. **f** Growth curve of tumor (one-sided ANOVA test, ***$P < 0.001$, $n = 5$). **g** Flow cytometric analysis and quantification of CD3⁺CD8⁺ cells (gate on single live cells) and Granzyme B expression and IFN-γ on CD8⁺ cells ($n = 5$). Data represent cumulative results from 1/2 independent experiments with 5 mice/group. The data are represented as mean ± SEM. Statistical differences between pairs of groups were determined by a two-tailed Student's t-test (ns present not significant, **$P < 0.01$, ***$P < 0.001$). Source data are provided as a Source Data file.

Furthermore, we revealed that the crosstalk between infiltrating macrophages and cancer cells promoted the CCL2 production by cancer cells, which in turn, led to sustained myeloid cell-mediated inflammation in residual tumors. In addition, we demonstrated that blocking CCR2 with specific antagonist might be a salvage option to inhibit the residual tumor and overcome the resistance of anti-PD-1 therapy.

Heat-ablated lesions are speculated to have three zones:[30] a central zone of high temperature (>60 °C), a transitional zone of sublethal hyperthermia (43–50 °C), and the surrounding tissue that

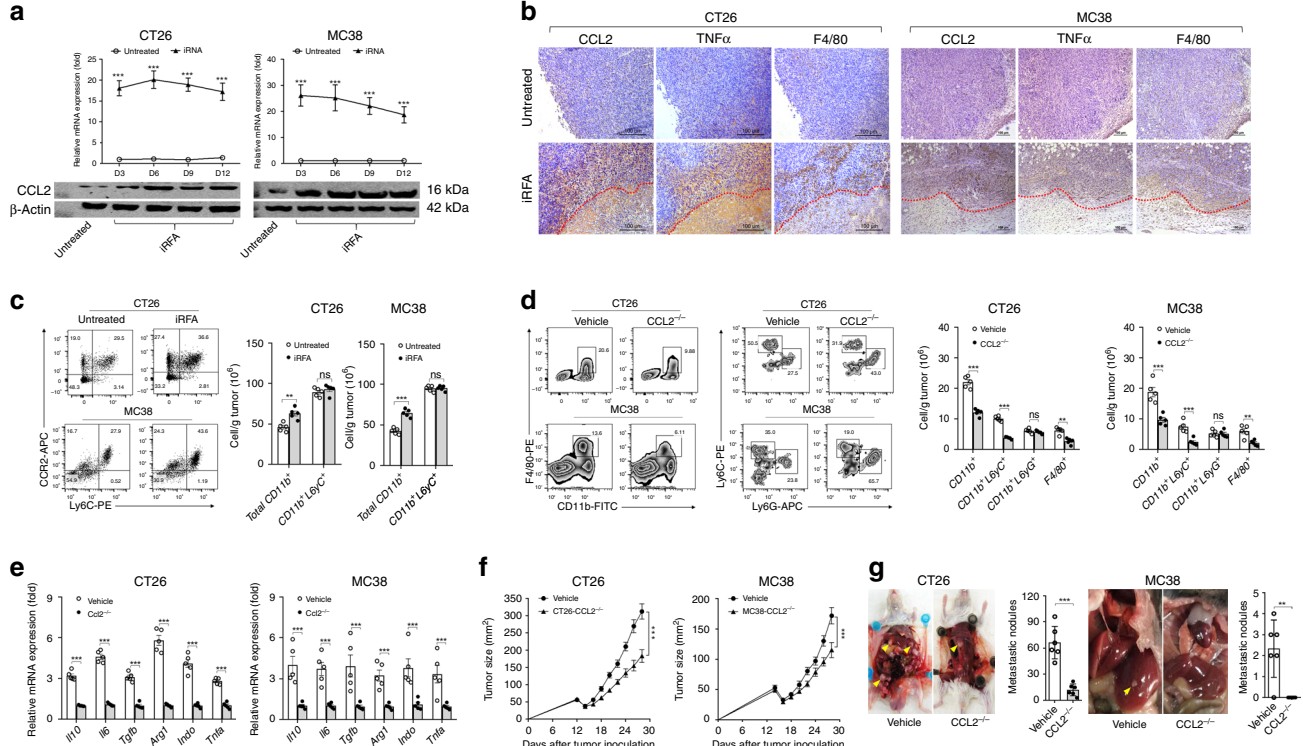

**Fig. 6 Tumor cell-derived CCL2 is critical for TAM accumulation in residual tumors. a** CCL2 mRNA and protein expression in the untreated and iRFA-treated CT26 and MC38 tumors on days 3, 6, 9, and 12 by real-time PCR and Western blot ($n = 5$). **b** Representative microphotographs are showing CCL2, TNFα, and F4/80 staining of the untreated and residual CT26 and MC38 tumor (the necrotic ablation area is below the red dotted line). Scale bar = 100 μm ($n = 3$). **c** Flow cytometric analysis and quantification of CCR2 expression on total myeloid cells and monocytes (gate on CD11b$^+$ cells) in the untreated and iRFA-treated CT26 and MC38 tumors on day 9 after iRFA ($n = 5$). **d** Flow cytometric analysis and quantification of CD11b$^+$, F4/80$^+$, Ly6G$^+$ and Ly6C$^+$cells in residual wild-type and CCL2$^{-/-}$ CT26 and MC38 tumors on day 9 after iRFA (gate on CD45$^+$ liver cells and CD11b$^+$ cells, respectively) ($n = 5$). **e** mRNA expression of IL-10, IL-6, TGFβ, Arg1, IDO, and TNFα in residual wild-type and CCL2$^{-/-}$ CT26 and MC38 tumors on day 9 after iRFA ($n = 5$). **f** The growth curve of residual wild-type and CCL2$^{-/-}$ CT26 and MC38 tumors treated with iRFA (one-sided ANOVA test, ***$P < 0.001$, $n = 5$). **g** The number of distant metastases in wild-type and CCL2$^{-/-}$ CT26 and MC38 tumor-bearing mice on day 14 after iRFA (yellow arrow indexes metastasis) ($n = 6$). Data represent cumulative results from 1/2 independent experiments. The data are represented as mean ± SEM. Statistical differences between pairs of groups were determined by a two-tailed Student's t-test (**$P < 0.01$, ***$P < 0.001$).

is unaffected by ablation. At the central zone, rapid protein denaturation occurs, which immediately leads to coagulative necrosis, while in the transitional zone, the tissues undergo sublethal injury. Thus, if the temperature high enough to kill tumor cells does not encompass the whole target tumor volume; the tumor cells in the transitional zone undergo reversible injury and eventually survive. This process can lead to rapid tumor progression[16,18,31–35]. Several mechanisms underlying this process, such as RFA-induced hypoxia microenvironment[32,34], activation of tumor-derived endothelial cells[31], epithelial-mesenchymal transition (EMT)[36,37], and autophagy[18], have been revealed. The current data obtained from RNAseq suggests that RFA elicits complex physiological reactions, including inflammatory response, immune response, response to stress, hypoxia, and wounding healing, ultimately leading to an inhibited anti-tumor immune response in residual tumors.

In ablated lesions, the proinflammatory cytokines are released from the ablated tissue or tumor and can trigger the release of additional cytokines and chemokines in the transitional zone, thereby leading to inflammation around the necrotic lesion[6]. In the study, we found RFA-elicited local inflammation was transient when the target tumor was ablated completely. The inflammation subsided with wound healing. However, the inflammatory response retained inside the residual tumors after iRFA was characterized by the infiltration of a large number of myeloid cells. Furthermore, we confirmed that tumor cells-

derived CCL2 is critical for the recruitment of monocytes and TAMs into tumors, which was consistent with the surgery-induced mobilization of myeloid cells[38]. Subsequently, the infiltrating macrophages promoted the residual tumor cells to produce CCL2 through TNFα. Thus, a positive feedback mechanism mediated by the crosstalk between the host and tumor cells is deduced, which facilitates the infiltration of monocytes and macrophages and leads to chronic inflammation as reported previously[25].

Tumor tissue myeloid cells play crucial roles in anti-tumor immunity. Monocytes, macrophages, and granulocytes are actively recruited to tumors and accelerate the tumor progression by suppressing the adaptive immune responses to tumor cells[39,40]. High infiltration of myeloid cells correlates with poor prognosis and immune checkpoint blockade resistance[41–43]. Targeting the migration of myeloid cells is a promising therapeutic strategy[26,44,45]. In a recent study, Liang et al. reported that treatment with anti-CCR2 mAbs blocks the infiltration of radiation-induced MDSC and alleviates the immunosuppression, following the activation of the STING pathway[46]. Dual-blockade CCR2 plus CXCR2 with specific antagonists reduces tumor-infiltrating myeloid cells and enhances chemotherapeutic responses in pancreatic adenocarcinoma patients and animal models[47]. In the present study, we showed that the CCR2 antagonist treatment alleviates the tumor progression after iRFA and reverses the resistance of residual tumors to PD-1 therapy.

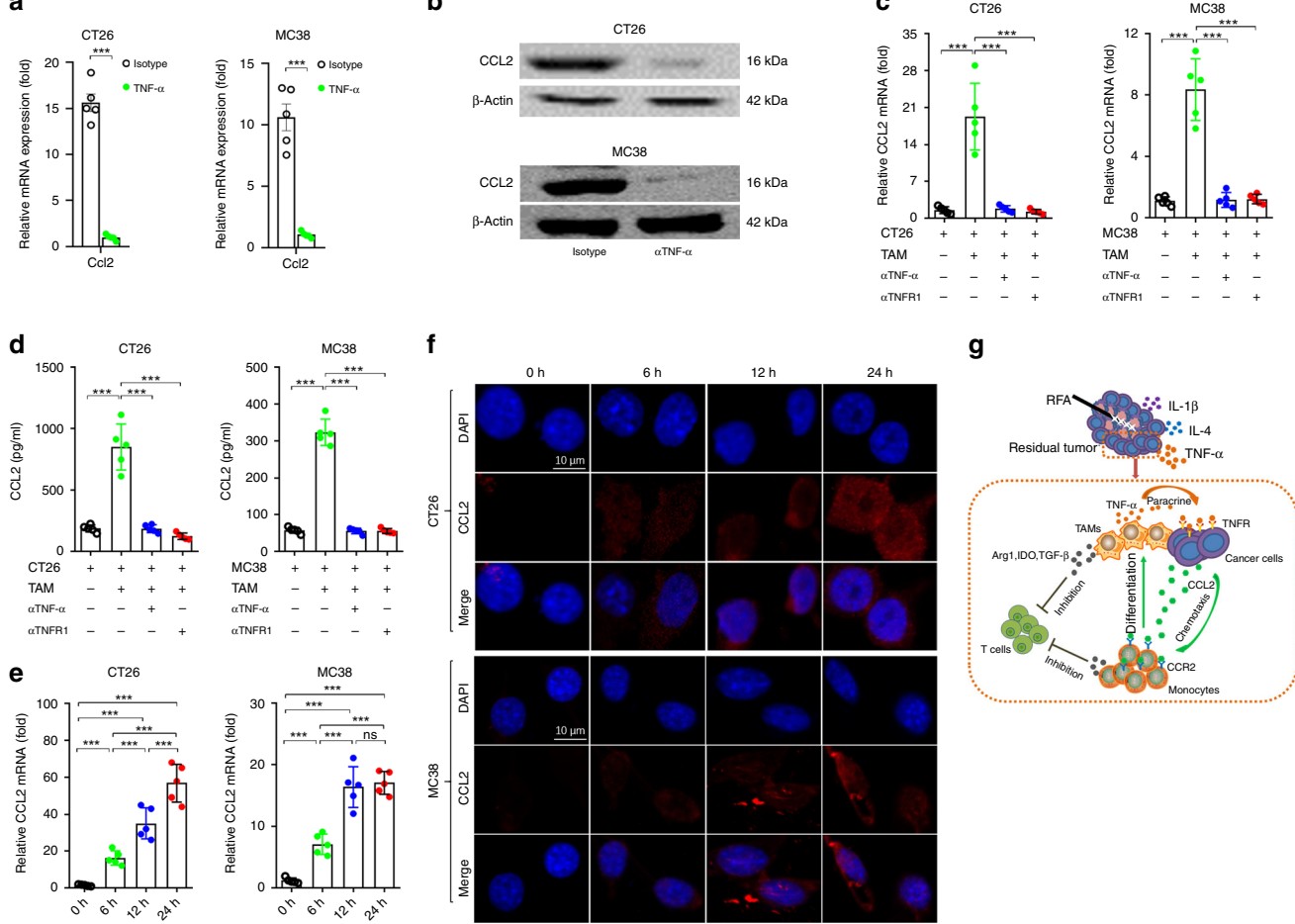

**Fig. 7** TAMs promote tumor cells to produce CCL2 through TNFα. **a, b** Mice were administered etanercept intraperitoneally before iRFA treatment. **a** CCL2 mRNA expression was quantified by real-time PCR on day 3 ($n = 5$). **b** CCL2 protein expressions were analyzed by Western blot on day 3 ($n = 5$). **c, d** CT26 or MC38 cancer cells were cocultured with TAMs (CD11b⁺F4/80⁺) isolated from residual tumors. To block TNFα and TNFR1, anti-mouse TNFα IgG (R&D Systems) or anti-TNFR1 mAb were added. **c** Total RNA was extracted, and *CCL2* mRNA was quantified by real-time PCR. Relative mRNA expression was expressed as fold-change ($n = 5$). **d** Concentration of secreted CCL2 protein in culture media was determined by ELISA ($n = 5$). **e, f** CT26 or MC38 cells were stimulated with recombinant mouse TNFα. **e** Relative mRNA expression was expressed as fold-change ($n = 3$). **f** CCL2 protein was detected by immunofluorescence analyses ($n = 3$). **g** Schematic depiction of the crosstalk between residual tumor cells and TAMs promotes CCL2 production by tumor cells, sustaining the infiltration of myeloid cells and inhibiting T cell activity in residual tumors. Data represent cumulative results from 1/2 independent experiments. The data are represented as mean ± SEM. Statistical differences between pairs of groups were determined by a two-tailed Student's *t*-test (***$P < 0.001$). Source data are provided as a Source Data file.

This finding provides a salvage alternative for residual tumors after iRFA. However, the treatment is not sufficient to eliminate the residual tumors and completely prevent the distant metastasis.

Despite continual advances in energy delivery and application technique, local recurrence remains one of the major disadvantages of thermal ablation, especially in patients with tumor larger than 3 cm, poorly defined margin, and large vessel proximity[48,49]. In the present study, 40 out of 43 patients with residual tumor had one or more risk factors. Despite multivariate analysis showed that iRFA was associated with earlier new metastasis and poor survival, it is hard to account for bias where the target lesions might be inherently more aggressive in contrast to the incomplete ablation leading to worse outcomes. This is a limitation of this study.

In summary, iRFA of a target tumor might lead to rapid tumor progression, thereby limiting the efficacy of CBI. Therefore, it is necessary to evaluate the risk factors associated with the local residual tumors, such as size > 3 cm, poorly defined margin, and large vessel and gastrointestinal tract proximity, in order to select

the patients who can benefit from the combined therapy of RFA and immunotherapy.

## Methods

**Patient selection and study.** A patient database was queried for CRCLM patients treated with RFA from January 2008 to December 2016 at the Third Affiliated Hospital, Soochow University. A total of 551 consecutive patients with were initially included in this study. Among them, 43 patients who presented with local residual tumor, which was defined as recurrent tumor within or at the periphery of the original ablative zone within 4 months after RFA (Group A). For the control group (Group B), 43 out of 508 patients, age- and sex-matched but without local residual tumor after RFA during the same study period were selected. Clinical data, including age, gender, location of primary tumor, time to liver metastasis, and carcinoembryonic antigen (CEA) level were obtained from the database. Baseline imaging evaluations of liver metastases were performed using enhanced CT and/or MRI. Tumor factors including size, number of lesions, adjacent relationship with large vessels, gastrointestinal tract and gallbladder were recorded. All patients underwent serial monitoring of CEA, CT and/or MRI at 2- to 3-month intervals for the detection of local residual tumor and new metastasis. Local residual tumor was defined as recurrent tumor within or at the periphery of the original ablative zone within 4 months after RFA.

RFA was performed percutaneously guided by ultrasonography (US) or computed tomography (CT). For tumors less than 3.0 cm in diameter, the

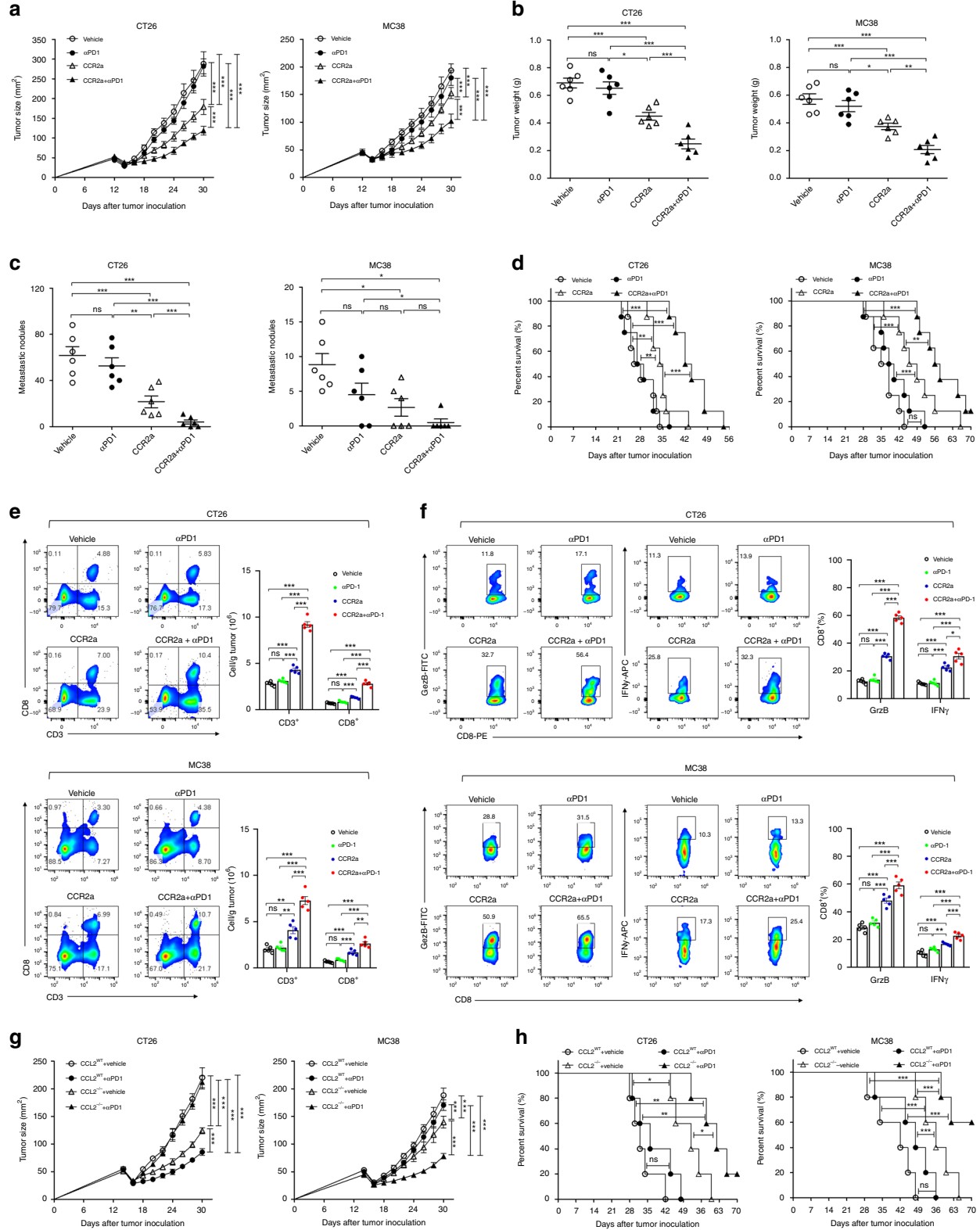

multitined expandable electrode (StarBurst XL, RITA) was deployed into the center of the tumor. Each application of RFA energy lasted for 15–25 min to gain a 5.0 cm ablation zone. For tumors larger than 3.0 cm, multiple overlapping zones of ablation were needed for the destruction of the tumor and a surrounding rim of the non-tumor liver. For patients with more than one lesion, the tumors were ablated separately.

This study was conducted according to the principles of the Declaration of Helsinki and was approved by the Ethics Committee of the Third Affiliated Hospital, Soochow University.

**Cell lines**. The mouse colon cancer cell line CT26 and MC38 were obtained from the Chinese Academy of Sciences, Shanghai Institute for Biological Sciences (Shanghai, China). Liver cancer cell line Hepa1-6 was purchase from Applied Biological Materials. Lnc (Zhenjiang, China).

**Animal models and treatments**. Balb/c (H2Kd) and C57BL/6 (B6; H2Kb) mice were purchased from Comparative Medicine Center, Yangzhou University. Balb/c-Tg (Tnfα-luc)-Xen (Caliper Lifescience) introduced by Model Animals Research Center,

**Fig. 8** CCL2/CCR2 blockade inhibits tumor progression and overcomes resistance to anti-PD-1 therapy. **a–f** iRFA treatment was performed in CT26 and MC38 colon cancer models as shown in Fig. 2a. Anti-PD-1 mAb (200 μg, clone: J43) was administered through intraperitoneal injection to mice every 3 days for a total of four times. The CCR2 antagonist (CCR2a) (RS504393, Tocris) was given subcutaneously at a dose of 5 mg/kg twice per day for 9 days. **a** Growth curve of the CT26 and MC38 residual tumor (one-sided ANOVA test, $n = 8$). **b** The weight of the residual CT26 and MC38 tumor examined on day 14 after iRFA by dissection of the mice ($n = 6$). **c** The number of metastases examined on day 14 after iRFA by dissection the mice ($n = 6$). **d** Kaplan–Meier survival curves are shown, and the log-rank test was performed ($n = 8$). **e** Flow cytometric analysis and quantification of CD3$^+$ and CD8$^+$ infiltration (gate on single live cells) in residual CT26 tumors. **f** Granzyme B and IFNγ expression on CD8$^+$ cells in residual CT26 tumors. (gate on CD8$^+$ cells) ($n = 5$). **g, h** iRFA treatment was performed in mice bearing wild type and CCL2$^{-/-}$ CT26 or MC38 tumor. **g** Growth curve of the CT26 and MC38 residual tumor (one-sided ANOVA test, $n = 5$). **h** Kaplan–Meier survival curves are shown, and the log-rank test was performed ($n = 8$). Data represent results from 1/2 independent experiments. The data are represented as mean ± SEM. Statistical differences between pairs of groups were determined by a two-tailed Student's t-test (ns present not significant, *$P < 0.05$, **$P < 0.01$, ***$P < 0.001$). Source data are provided as a Source Data file.

---

Nanjing University. A density of $1 \times 10^6$ CT26, MC38 or Hepa1-6 cells were injected i.d. into male BALB/C and C57BL/6 mice on the right flank, respectively. The treatments were initiated when the longest diameter reached about 0.8 cm. RFA was performed using a 17-gauge single ablation electrode (RITA Medical Systems Inc., Mountain View, CA, USA) with 1 cm active tip inserted percutaneously. The ablation power was set at 7 Walt. Complete ablation: electrode was punctured along the long axis of the tumor, and the tip of the electrode reached the contralateral edge of the tumor. Treatments were administered for 3.5–4.5 min at the target temperature of 70 °C to ensure complete ablation of the target tumors. Partial ablation: the tip of the electrode reached the middle point of the tumor's long axis. The treatments were administered for 1.5–2.0 min at the target temperature of 70 °C for partial necrosis of the target tumors. For PD-1 blockade, 200 μg anti-PD-1 (clone: J43, catalogue: BE0033-2, BioXcell) was administered through intraperitoneal injection to mice every 3 days for a total of four times. The CCR2 antagonist (RS504393, Tocris) was given subcutaneously at a dose of 5 mg/kg twice per day for 9 days. To neutralize TNFα in vivo, 200 μg (10 mg/kg) etanercept (Pfizer, New York, NY, USA) was injected intraperitoneally. The perpendicular diameters of the tumor were measured using calipers every 2 days. The tumor size was calculated using the formula, $L \times W$, where $L$ is the longest diameter and $W$ is the perpendicular dimension.

All experimental protocols were approved by the Committee for the Protection of Animal Care Committee at Soochow University and Central South University. Animal testing and research conformed to all relevant ethical regulations.

**Flow cytometric analysis.** The tumor masses were removed, homogenized, and digested with collagenase and hyaluronidase solution. The resulting cell suspension was filtered through a cell mesh and resuspended in Hank's media plus 1% fetal calf serum (FCS) for further analysis. Antibodies to CD45 (30-F11, dilution 1:200, Cat 25045182), CD3 (145-2c11, dilution 1:100, Cat 15003842), CD4 (GK1.5, dilution 1:200, Cat 11004181), CD8 (53-6.7, dilution 1:100, Cat MA1-10304), granzyme B (NGZB, dilution 1:200, Cat 11889882), IFN-γ (XMG1.2, dilution 1:400, Cat 17731181), FoxP3 (FJK-16S, dilution 1:100, Cat 12577382), F4/80 (BM8, dilution 1:100, Cat MF480043), CD11b (M1/70, dilution 1:200, Cat 11011282), Ly6G (1A8, dilution 1:200, Cat 17966882), CD206 (19.2, 1:100, Cat 56206942), MHC-II (M5/114.15.2, dilution 1:200, Cat 46532182), Gr1 (RB6-8C5, dilution 1:100, Cat MA1-83934), PD-L1 (MIH5, dilution 1:100, Cat 17598282) and PD-1 (RPM1-30, dilution 1:100, Cat 17998182) were purchased from eBioscience. Antibody to Ly6C (AL-21, dilution 1:200, Cat 560595) was purchased from BD Biosciences. Antibody to CCR2 (475301, dilution 1:200, Cat FAB5538P) was purchased from R&D Systems. For intracellular cytokine staining, the cells were permeabilized using a FoxP3 Fixation and Permeabilization Kit (eBioscience) and stained for Foxp3. For intracellular cytokine staining, the harvested cells were stimulated with PMA (50 ng/mL) and ionomycin (500 ng/mL) for 4 h and incubated for 1 h with brefeldin A (10 μg/mL). Subsequently, flow cytometric analysis was performed using a FACS flow cytometer (Canto II, BD), and the data were analyzed using FlowJo software (Treestar).

**Total RNA extraction, real-time PCR, and RNA-seq.** Total RNA was extracted from tissues, using a total RNA purification kit (Shenergy Biocolor BioScience & Technology Company, Shanghai, China), according to the manufacturer's instructions. An equivalent of 2 μg total RNA was reverse transcribed into cDNA using the first strand cDNA synthesis kit (Fermantas, Vilnius, Lithuania), according to the manufacturer's instructions. The primers, TaqMan probes, and the reference gene β-actin were designed according to the National Center for Biotechnology Information (NCBI) database using the Primer Premier 5.0 software (Palo Alto, CA, USA). The mRNA levels of the target genes and reference gene β-actin were measured using a real-time PCR machine, ABI 7500 (Applied Biosystems, USA). The mRNA levels of the target genes were normalized to that of Gapdh ($\Delta C_t = C_t$ gene of interest–$C_t$ Gapdh) and reported as relative mRNA expression fold-change. RNA-seq was performed using CapitalBio Technology (Beijing, China), and the data were expressed as mean displayed in the center of the heatmaps. The fold-change was calculated and converted to log$_2$.

**IHC procedures.** Four micrometers thick tumor tissues were fixed in 10% formaldehyde, embedded in paraffin, dewaxed in xylene, and rehydrated through a graded series of ethanol. Then, tumor tissue sections were subjected to routine hematoxylin-eosin (H&E), and IHC staining by CD11b (Cat ab133357, 1:500, Abcam), CD31 (Cat ab28364, 1:1000, Abcam), F4/80 (D2S9R) (Cat 70076S, 1:500, XP), CCL2 (Cat ab25124, 1:1000, Abcam) and TNFα (Cat ab6671, 1:1000, Abcam) staining. Briefly, the sections were incubated overnight with primary antibody, followed by incubation with the biotinylated antibody (1:2000, anti-rabbit, Sigma–Aldrich). Subsequently, the slides were washed with PBS and treated with diaminobenzidine (DAB) chromogen for 3–5 min that served as the chromogen, and hematoxylin was used for the nuclear counterstain. Then, the sections were dehydrated, cleared, and mounted. The H&E and IHC staining images were obtained using a microcamera (Leica TCS SP8 MP, Germany).

**Isolation of myeloid or CD8$^+$ T cells from tumors or spleen.** The spleens and tumor masses were removed, homogenized, and digested with collagenase and hyaluronidase solution (2.5 mg/mL collagenase I, 1 mg/mL collagenase IV, 0.25 mg/mL hyaluronidase IV-S, 300 μg/mL DNase I, and 0.01% HEPES in RPMI1640) at 37 °C for 2 h. Then, the pieces were gently pressed between the frosted edges of two sterile glass slides, and the cell suspension was filtered through a 40-μm cell strainer to remove the debris and cell clumps. The single cell suspension was washed and resuspended in Hank's Balanced Salt (HBSS). Then, the cells were stained with anti-CD45, anti-CD11b, and anti-CD8 antibodies for sorting on a FACS flow cytometer (Aria II, BD). The dead cells were excluded using DAPI (Invitrogen).

**T cell suppression assay.** The isolated splenic CD8$^+$ cells were labeled with 1 mM CFSE (Cat 65085084, Invitrogen) in pre-warmed PBS for 10 min at 37 °C. Then, the CFSE-labeled CD8$^+$ T cells were plated in complete RPMI media supplemented with 0.05 M β-mercaptoethanol into 96-well plates ($25 \times 10^3$ cells/well) coated with 1 μg/mL anti-CD3 (145-2C11, Cat 100340, Biolegend) and 5 μg/mL anti-CD28 (37.51, Cat 102116, Biolegend) antibodies. The myeloid cells isolated from spleens or residual tumors were added (1:1) to the wells and incubated at 37 °C. After 48 h, the cells were harvested and CFSE signal in the gated CD8$^+$ T cells was measured by flow cytometry (Canto II, BD).

**Bioluminescence imaging.** Bioluminescence imaging (Lumina II, Caliper Life Sciences) was used to analyze the TNFα expression before and at 3, 6, 9 days after RFA treatment. A volume of 200 μL D-luciferin (Cat 119222, Caliper Lifescience) (10 μg/g body weight) was injected intraperitoneally into BALB/C-Tg (Tnfa-luc)-Xen mice. After 10 min, the mice were anesthetized with isoflurane. Finally, the mice were photographed in the loading room (5 min imaging), and the images were saved. The expression of TNFα protein was quantified using Living Image software (Living Image 4.3.1, Caliper Life Sciences).

**Western blot analysis.** An equivalent of 30–50 μg total cellular protein was separated on 4–20% gradient SDS-PAGE (Bio-Rad Laboratories). The proteins were transferred to nitrocellulose membrane (Pall Corporation, Ann Arbor, MI, USA), and the membranes were blocked with 0.1% casein in TBS for 60 min Then, the blots were probed in 0.1% casein/TBS-T with CCL2 mouse mAb (dilution 1:1000, Cat 66272-2-Ig, Proteintech) overnight at 4 °C. Subsequently, the blots were incubated with infrared-labeled secondary Abs at 1:10,000 at room temperature for 1 h. The immunoreactive bands were visualized and quantified using a LI-COR Odyssey CLx infrared imaging system (LI-COR Biosciences, Lincoln, NE, USA).

**Measurements of cytokines and chemokine in peripheral blood.** Venous blood was obtained from the orbit of mice 9 days after RFA. The samples were clot for 30 min, then centrifuge for 10 min at $1000 \times g$. Remove serum and assay immediately using ELISA kit specific for mouse IL-6 (Cat 550950, BD Biosciences), IL-10 (Cat 555252, BD Biosciences), TGFβ (Cat 555052, BD Biosciences) and CCL2 (Cat SMZE00B, R&D Systems).

**CCL2 gene knockout in CT26 and MC38 cancer cells**. The CCL2 gene was knocked out by CRISP/Cas9 editing technique in CT26 and MC38 cancer cells. sgRNA design and Cas9 vector construction were performed by Applied Biological Materials. Lnc (Zhenjiang, China). In brief, sgRNAs were designed against mCCL2 gene (Mus musculus, NM_144500.4) using CHOPCHOP (https://chopchop.rc.fas. harvard.edu/). Lentivirus packaging was performed and transduced to CT26 and MC38 cells according to abm protocol (https://www.abmgood.com/Lentivirus-Packaging-Systems.html). Puromycin selection was performed two days after transduction.

**Cancer cells stimulation and CCL2 measurements**. CT26 mouse colon cancer cells were plated at a density of $1 \times 10^5$ cells/well in 12-well plates. The cells were stimulated with recombinant mouse TNFα (Cat 410-MT-050, R&D Systems) (10 ng/mL) or macrophages ($CD11b^+F4/80^+$ isolated from residual tumors) at 37 °C for 6–48 h. To block TNFα and TNFR1, 2 μg anti-mouse TNFα IgG (Cat MAB4101, R&D Systems) or anti-TNFR1 mAb (Cat MAB430-100, R&D Systems) was added to each well. The level of CCL2 secreted protein was measured in the cell supernatants using an ELISA kit specific for mouse CCL2 (Cat SMZE00B, R&D Systems), following the manufacturer's instructions.

**Immunofluorescence analyses**. $1 \times 10^5$ CT26 cancer cells seeded on coverslips in 6-well plates and cultured to 70–80% confluency and cultured for 24 h. Then, the cells were fixed in 4% paraformaldehyde for 15 min, blocked with 5% BSA for 30 min, incubated with the anti-CCL2 antibody (ab25124,1:400, Abcam) overnight at 4 °C, followed by incubation with secondary antibodies (1:3000, Jackson). To visualize the nucleus, the cells were stained with DAPI (Sigma). The fluorescence images were observed under a laser-scanning confocal microscope (Carl Zeiss, Germany).

**Statistical analyses**. Differences in the distribution of selected demographic and clinical characteristics between RFA and non-RFA groups were evaluated using Student's t-test and Fisher's chi-square test. The time to new metastasis (TTNM) was calculated from the date of RFA to the date of detection of new metastasis (including extrahepatic and intrahepatic metastases separated from the RFA zone). OS was calculated from the date of RFA to the date of death from any cause or the date of the last follow-up. The OS and TTNM curves were estimated using the Kaplan–Meier method and compared using the log-rank test. Retrospective stratification according to the Cox proportional hazards regression model was used for the adjustment of the putative underlying risk factors in earlier new metastasis. These factors were evaluated with univariate analyses using Cox regression tests. All clinical variables that were statistically significant in the univariate analysis were included in a multivariate analysis. $P < 0.05$ was considered statistically significant.

Data from animal experiments were expressed as mean ± SEM (standard error of mean) for biological replicates and mean ± SD (standard deviation) for technical replicates. Two-tailed unpaired Student's t-test was used for the comparison of RFA-treated mice and control groups. ANOVA test was used for the comparison of the tumor growth curves. Survival data were analyzed by log-rank test. $P < 0.05$ was considered as statistically significant. Data were analyzed using SPSS software (Version 13.0, SPSS Inc., Chicago, IL, USA) and GraphPad Prism (Version 8.0, GraphPad Software, SanDiego, California USA).

**Reporting summary**. Further information on research design is available in the Nature Research Reporting Summary linked to this article.

## Data availability

All relevant data related to this manuscript are available on request from the authors on reasonable request. The accession number for the RNA-sequencing data described in this study is GSE138224. Original un-cropped western blots are provided in source data. The source data underlying Figs. 1, 2, 4–8, Supplementary Table 3 and Supplementary Figs. 1, 2, 4–9 are provided.

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

## Acknowledgements

This work was supported by grants from the National Natural Science Foundation of China (NSFC) 81773234, 81201741 (to L.R.S), 81671676 (to W.H.L.), 31570908 (to C.P. W.), 18KJA180011 (Y.B.Z), 81773749 (to Y.Z). Scientific Research Project of Hunan Health and Health Commission C2019189 (to L.R.S.).

## Author contributions

L.S., W.L., Y.-B.Z., C.W., L.Z. and W.W. designed research studies and analyzed data. L.S. wrote the manuscript. L.S., J.W., X.-Y.L. and H.S. performed RFA treatments in CRCLM patients and collected clinical data. L.S., J.W., Y.Z, N.D., D.L., S.D., X.W., C.P., H.S., X.-D.L. and C.Z. conducted experiments. W.L. and C.W. provided guidance for the research. All authors were involved in the drafting, review, and approval of the report and the decision to submit for publication.

## Competing interests

The authors declare no competing interests.
