## [Peer Review File · Nature Communications]

Reviewers' comments:

Reviewer #1 (Remarks to the Author):

This is an intriguing study that suggests that incomplete local treatment in the form of RFA may give rise to distant metastases and systemic tumor progression due to upregulated antitumor immunity. This is suggested clinically using a retrospective controlled study involving more than 500 treated tumors where incomplete ablation was found to be an independent risk factor for earlier development of metastases and inferior survival. In corresponding animal models, iRFA was associated with local resistance to future treatment with PD-1 inhibitors, and interestingly also associated with systemic resistance to PD-1 therapy and the development of distant metastases as well. There were complex genetic changes in the incompletely treated primary tumor reflecting inflammatory changes led to an influx of inhibitory cells. This recruitment of TAM was particularly dependent on CCL2 induced by TNFalpha, which are identified as potential therapeutic targets in this setting.

Overall this is a provocative and well done translational study with potential clinical and therapeutic implications, and is therefore worthy of publication. However, major issues include the need to more conclusively demonstrate that inhibitory mechanisms demonstrated are specifically related to iRFA and not also present in the setting of cRFA. All experiments except for PD-1/CCR2 inhibition also seem to have been performed on CT26 and therefore it is hard to know whether these results are generalizable to other model systems. The manuscript would also benefit from more careful grammatical editing throughout. Other more specific comments are below.

Introduction

- Reference 12 is not properly cited and is an important part of the rationale for this study
- It would be helpful to more specifically describe findings from previous studies in terms of iRFA leading to progression within or outside of the treated lesions given the current study findings.

Results

- Despite the authors use of a MVA, it is hard to account for bias where the iRFA lesions are inherently more aggressive in contrast to the incomplete ablation leading to worse outcomes – this should be mentioned as a limitation in the discussion.
- As mentioned above, more comparisons with cRFA need to be performed to demonstrate the inhibitory effects are not a result of RFA in general – for example, does cRFA also cause increase in MDSC in the peripheral blood?
- Is there any additional benefit to PD-1 inhibition in the setting of iRFA and CCL2 knock out with CRISPR?

Method

- It should be noted whether local response in the iRFA lesions was determined blinded to the development of distant progression, which would have been understandably difficult. However, if this was not blinded there is the potential for bias.
- There are several typos in the methods – e.g. “presented local tumor”, “baseline imaging evaluations liver metastases”
- How were the treatment parameters chosen for partial ablation? Were other parameters tried? Was there a minimum threshold below which the suppressive effects were not observed?

Reviewer #2 (Remarks to the Author):

Major findings:

Retrospective review of patient CT's showed that patients with residual metastatic to the liver after

RFA was an independent risk factor for earlier metastasis and poor survival. Using mouse models they show incomplete RFA of cancers was associated with immune suppressive myeloid cells that impaired T cell function and is thought to be driven by CCL2. The strength of the mechanistic findings are considerable given a CCR2 antagonist could restore some T cell function.

How do the authors resolve these findings with the statements they make in the introduction about "the synergistic

54 antitumor effects of RFA in combination with anti-PD-1 therapy in a preclinical
55 study 11. Recently, a phase II clinical trial showed that local ablation combined
56 with anti-PD-1 therapy significantly improves the survival of patients with
57 oligometastatic non-small cell lung cancer (NSCLC)>"

This is a well done study, clearly presented and tells an interesting story. The reliance on a single model of CRC is a weakness. There are otherwise no major deficiencies.

Sincerely,
Robert A. Anders
Baltimore MD 3/2019

Point-to-point answers to reviewer comments

All changes in the text are in red color.

Reviewer #1

This is an intriguing study that suggests that incomplete local treatment in the form of RFA may give rise to distant metastases and systemic tumor progression due to upregulated antitumor immunity. This is suggested clinically using a retrospective controlled study involving more than 500 treated tumors where incomplete ablation was found to be an independent risk factor for earlier development of metastases and inferior survival. In corresponding animal models, iRFA was associated with local resistance to future treatment with PD-1 inhibitors, and interestingly also associated with systemic resistance to PD-1 therapy and the development of distant metastases as well. There were complex genetic changes in the incompletely treated primary tumor reflecting inflammatory changes led to an influx of inhibitory cells. This recruitment of TAM was particularly dependent on CCL2 induced by TNFalpha, which are identified as potential therapeutic targets in this setting.

Overall this is a provocative and well done translational study with potential clinical and therapeutic implications, and is therefore worthy of publication. However, major issues include the need to (1) more conclusively demonstrate that inhibitory mechanisms demonstrated are specifically related to iRFA and not also present in the setting of cRFA. (2) All experiments except for PD-1/CCR2 inhibition also seem to have been performed on CT26 and therefore it is hard to know whether these results are generalizable to other model systems. (3) The manuscript would also benefit from more careful grammatical editing throughout. Other more specific comments are below.

Introduction

(4)- Reference 12 is not properly cited and is an important part of the rationale for this study. It would be helpful to more specifically describe findings from previous studies in terms of iRFA leading to progression within or outside of the treated lesions given the current study findings.

Results

(5) - Despite the authors use of a MVA, it is hard to account for bias where the iRFA lesions are inherently more aggressive in contrast to the incomplete ablation leading to worse outcomes – this should be mentioned as a limitation in the discussion.

(1) As mentioned above, more comparisons with cRFA need to be performed to demonstrate the inhibitory effects are not a result of RFA in general – for example, does cRFA also cause increase in MDSC in the peripheral blood?

(6)- Is there any additional benefit to PD-1 inhibition in the setting of iRFA and CCL2 knock out with CRISPR?

Method

(7)- It should be noted whether local response in the iRFA lesions was determined blinded to the development of distant progression, which would have been understandably difficult. However, if this was not blinded there is the potential for bias.

(8)- There are several typos in the methods – e.g. “presented local tumor”, “baseline imaging evaluations liver metastases”

(9)- How were the treatment parameters chosen for partial ablation? Were other parameters tried? (10) Was there a minimum threshold below which the suppressive effects were not observed?

Comment 1: more conclusively demonstrate that inhibitory mechanisms demonstrated are specifically related to iRFA and not also present in the setting of cRFA.

Response: We performed additional experiments to examine the peripheral MDSCs, immunosuppressive cytokines IL-10 and TGF- β and chemokine CCL2 protein in CT26 and MC38 models after incomplete radiofrequency (iRFA) and complete RFA (cRFA). We found the number of MDSCs was reduced and the concentration of IL-10, TGF- β and CCL2 was decreased in mice treated with cRFA as compared to iRFA and untreated mice. These data are shown in new Fig S5.

Comment 2: All experiments except for PD-1/CCR2 inhibition also seem to have been performed on CT26 and therefore it is hard to know whether these results are generalizable to other model systems.

Response: To extend the previous data, we performed additional experiments using another colon cancer model MC38. In general, the new data confirmed the previous findings in CT26 and hepa1-6 models. The new data now are presented in parallel with the previous data obtained from CT26 model in the revised Fig 2, 4-8, Fig S1, 2, 4-6.

Comment 3: The manuscript would also benefit from more careful grammatical editing throughout.

Response: According to the Reviewer's suggestion, we have asked a native English speaker to check grammatical editing.

Comment 4: Reference 12 is not properly cited and is an important part of the rationale for this study. It would be helpful to more specifically describe findings from previous studies in terms of iRFA leading to progression within or outside of the treated lesions given the current study findings

Response: According to the Reviewer's suggestion, we removed the reference and added two reference in terms of rapid tumor progression after iRFA.

Comment 5: Despite the authors use of a MVA, it is hard to account for bias where the iRFA lesions are inherently more aggressive in contrast to the incomplete ablation leading to worse outcomes – this should be mentioned as a limitation in the discussion.

Response: We agree that the target lesions might be inherently more aggressive and lead to worse outcomes. We have discussed about this point in the revised manuscript and we have updated the references accordingly (page 12).

Comment 6: Is there any additional benefit to PD-1 inhibition in the setting of iRFA and CCL2 knock out with CRISPR?

Response: Thank you for this insightful suggestion. We established iRFA-treated model with wild-type and CCL2^{KO} CT26 and MC38 cell lines, and found PD-1 therapy was effective to further inhibit the growth of residual CCL2^{KO} tumor and prolong the survival time after iRFA. New data are showed in Fig 8G, H.

Comment 7: It should be noted whether local response in the iRFA lesions was determined blinded to the development of distant progression, which would have been understandably difficult. However, if this was not blinded there is the potential for bias.

Response: Due to nature of RFA treatment, it is difficult to use blind method to evaluate local response after treatment. When designing this study, we realized the potential bias. So, we also compared the weight of local tumors examined on day 14 after iRFA by dissection the mice. This may be helpful to reduce bias.

Comment 8: There are several typos in the methods – e.g. “presented local tumor”, “baseline imaging evaluations liver metastases”

Response: We thank this reviewer for kindly pointing out these errors. We have carefully checked typos in the revised manuscript.

Comment 9: How were the treatment parameters chosen for partial ablation? Were other parameters tried?

Response: In the study, we control the ablation area by adjusting the position of electrode tip position and ablation time. In preliminary experiments, we established the parameters of cRFA and iRFA in subcutaneous transplanted tumors of mice using a 17-gauge single ablation electrode (RITA Medical Systems). As shown in the Fig 1, electrode was punctured along the long axis of the tumor, and the tip of the electrode reached the contralateral edge of the tumor to achieve complete ablation. For iRFA, the tip of the electrode reaches the middle point of the tumor's long axis. We did not try other parameters for partial ablation in this study. In a previous study, infrared thermal imaging was used to monitor the temperature of ablation area. (Z. Zhao et al. Cancer Letters, 2018). We think this may be a more precise method to control the extent of ablation, and it is worth trying in future studies.

Figure 1. Treatment parameters for complete and partial ablation

Comment 10: Was there a minimum threshold below which the suppressive effects were not observed?

Response: This is a critical issue worthy of in-depth discussion. Further studies are warranted, as it not only is helpful to find new strategies to prevent the recurrence of ablated tumors, but also may provide new insights into the relationship between inflammation and tumor progression. Therefore, we wish to explore an experimental model for precise control of residual tumors, and ultimately understand the threshold for initiating immunosuppression.

Reviewer #2 :

Major findings: Retrospective review of patient CT's showed that patients with residual metastatic to the liver after RFA was an independent risk factor for earlier metastasis and poor survival. Using mouse models they show incomplete RFA of cancers was associated with immune suppressive myeloid cells that impaired T cell function and is thought to be driven by CCL2. The strength of the mechanistic findings are considerable given a CCR2 antagonist could restore some T cell function.

(1) How do the authors resolve these findings with the statements they make in the introduction about "the synergistic antitumor effects of RFA in combination with anti-PD-1 therapy in a preclinical study? Recently, a phase II clinical trial showed that local ablation combined 56 with anti-PD-1 therapy significantly improves the survival of patients with oligometastatic non-small cell lung cancer (NSCLC)" .

This is a well done study, clearly presented and tells an interesting story. (2) The reliance on a single model of CRC is a weakness. There are otherwise no major deficiencies.

Comment 1: How do the authors resolve these findings with the statements they make in the introduction about "the synergistic antitumor effects of RFA in combination with anti-PD-1 therapy in a preclinical study?"

Response: The present study was aimed to evaluate the immune response in residual tumors after incomplete RFA (iRFA). We have deleted the inappropriate statement in the section of introduction in the revised manuscript, and more specifically describe findings from previous studies in terms of iRFA leading to progression.

Comment 2: The reliance on a single model of CRC is a weakness.

Response: We performed additional experiment using another colon cancer model MC38. In general, the new data confirmed the previous finding in CT26 and hepa1-6 models. The new data now are presented in parallel with the previous data obtained from CT26 model in the revised Fig 2, 4-8, Fig S1, 2, 4-6.

REVIEWERS' COMMENTS:

Reviewer #1 (Remarks to the Author):

The authors have responded well to the prior comments. I would encourage them to either allow their responses to be viewed as a supplemental file or add responses to 9 and 10 to the manuscript / supplemental data.

Reviewer #2 (Remarks to the Author):

Important topic that needs more attention. This is an interesting model that might agree with a similar experience in humans.